# In Vivo Assessment of the Effects of Mono-Carrier Encapsulated Fucoxanthin Nanoparticles on Type 2 Diabetic C57 Mice and Their Oxidative Stress

**DOI:** 10.3390/antiox11101976

**Published:** 2022-10-02

**Authors:** Xin Zhang, Minghao Fan, Ke Luo, Wanying Xu, Jianjun Dong, Dongfeng Wang, Lu Chen, Junhong Yu

**Affiliations:** 1State Key Laboratory of Biological Fermentation Engineering of Beer, Tsingtao Brewery Co., Ltd., Qingdao 266100, China; 2College of Food Science and Engineering, Ocean University of China, Qingdao 266003, China

**Keywords:** fucoxanthin, zein, nanoparticles, T2DM, C57, diabetes

## Abstract

Fucoxanthin (FX) is a carotenoid from a marine origin that has an important role in our health, especially in the regulation and alleviation of type 2 diabetes. Its specific molecular structure makes it very unstable, which greatly affects its delivery in the body. In this study, FX was encapsulated in a mono-carrier using a hydrolyzed zein to form a nanocomplex with a stable structure and chemical properties (FZNP). Its stability was demonstrated by characterization and the efficacy of FX before and after encapsulation in alleviating diabetes in mice, which was evaluated by in vivo experiments. FZNP reduced the level of fasting blood glucose and restored it to normal levels in T2DM mice, which was not caused by a decrease in food intake, and effectively reduced oxidative stress in the organism. Both FX and FZNP repaired the hepatocyte and pancreatic β-cell damage, increased serum SOD and reduced INS values significantly, upregulated PI3K-AKT genes as well as CaMK and GNAs expression in the pancreas. FZNP increased ADPN and GSH-PX values more significantly and it decreased serum HOMA-IR and MDA values, upregulated GLUT2 expression, promoted glucose transport in pancreatic and hepatocytes, regulated glucose metabolism and glycogen synthesis with much superior effects than FX.

## 1. Introduction

Diabetes is a global disease that can lead to chronic complications such as cardiovascular disease, renal dysfunction, amputation and loss of vision [1]. Diabetes is usually classified as, type 1 diabetes mellitus (T1DM), type 2 diabetes mellitus (T2DM) and other types of diabetes. T1DM is a type of diabetes caused by the immune system. T2DM is a chronic disease characterized by hyperglycemia due to damaged β-cell activity, insulin resistance and impaired hepatic glycogen synthesis, and it accounts for 90–95% of patients with diabetes [2]. Patients with T2DM have reduced insulin levels within half an hour after a meal, which can lead to elevated blood glucose. The resulting hyperglycemia can lead to β-cell exhaustion and oxidative stress, which in turn can lead to β-cell death, a condition known as glucotoxicity [3]. Glucotoxicity can eventually lead to insulin resistance, which is the inability of insulin target tissues to respond effectively to insulin [4]. Many antidiabetic drugs are currently available for the treatment of T2DM, such as metformin, glyburide and thiazolidinedione, and the long-term use of these chemical drugs may cause cytotoxicity and adverse physical side effects [5]. As a result, research into the use of natural dietary supplements to treat and alleviate these chronic conditions has attracted increasing scientific attention [6].

Fucoxanthin (FX), a marine “gold” commonly found in macro algae and microalgae, is the most abundant carotenoid in nature, accounting for more than 10% of the total natural carotenoid production [7]. FX has received widespread attention in recent years because of its many physiological activities, namely antioxidant properties, anti-inflammatory, prevention of non-alcoholic fatty liver disease, hypolipidemic, anti-obesity, anti-diabetic and anti-cancer effects [8,9,10]. For example, it is employed as a chemotherapeutic carotenoid in the treatment of colon cancer [11]. FX, as the “gold” from the sea, is a promising candidate for the future treatment of human diseases and for development as a functional food or dietary supplement. It has anti-diabetic effects by regulating insulin signaling pathways [12]. Studies have demonstrated that FX significantly reduces plasma glucose levels and glucose intolerance as well as reducing the expression of pro-inflammatory genes [9,13], it not only increases insulin receptor expression but also improves insulin sensitivity in mice and helps to improve insulin resistance [14]. Like many carotenoids, FX is insoluble in water and susceptible to environmental influences, which greatly affects its delivery and effect in the body. Thanks to the study of targeted delivery systems, a new way of in vivo delivery of the active substance has been made possible. Targeted delivery systems are now at the forefront of research in order to effectively deliver active ingredients to the body. Various drug delivery systems with an excellent ability to delay drug release directly to the intestinal region have been investigated and developed, including natural, synthetic and semi-synthetic polymers as carriers in various delivery forms such as microspheres, nanoparticles and particles, liposomes, microcapsules and microtablets to overcome the intense acidic and enzymatic environment of the upper gastrointestinal tract [15].

Zein, a mixture of maize proteins with an average molecular weight of 25,000 to 45,000, is an excellent material for the fabrication of functional colloidal particles and for the controlled release of drugs and bioactive compounds [16]. It is insoluble in water and can be self-assembled by an anti-solvent method to form nanoparticles for the delivery of various active substances such as curcumin [17], lignin [18], insulin [19], quercetin [20] and resveratrol [21]. Zein colloidal particles are negatively charged at the surface and tend to lose physical stability as they approach neutral pH, which can lead to particle aggregation, and therefore using zein for encapsulation of active substances must be accompanied by another protein or two proteins or polysaccharides for protection [22]. Zein hydrolysate (ZH), a short-chain peptide produced by the hydrolysis of zein, which can also self-assemble in water to form nanoparticles, could theoretically encapsulate active substances in the same way as zein. However, very little research has been conducted on such peptides as carriers [23]. ZH not only can be used as a mono-carrier wall material to prevent nanoparticles from aggregating near the isoelectric point of the protein, but also its raw material has an industrial scale of production, which saves production costs, reduces the use of additives and the burden on the intestine, and is an excellent wall material for encapsulating active substances in the future. Therefore, in this experiment, we employed for the first time a colloidal nanoparticle-based system consisting of a mono-carrier of ZH encapsulated FX and verified its stability by physicochemical indicators. The function of the encapsulated FX in alleviating T2DM disease was evaluated through a 15-week intervention in a C57 mouse model of diabetes, compared with free FX to elucidate the differences in their antidiabetic efficacy, and to further investigate the mechanisms and key signaling pathways of FX intervention in T2DM.

## 2. Materials and Methods

### 2.1. Materials and Chemicals

Zein (>92%, Z3625), fucoxanthin (>95% purity, F6932) and Alcalase^®^ (2.4 L, P4860) were purchased from Sigma-Aldrich (St. Louis, MO, USA). Metformin Hydrochloride was from Bristol Myers Squibb Co. Ltd. (Shanghai, China). TRIzol reagents were purchased from Thermo Fisher Scientific (Franklin, MA, USA). The SPARKscript II RT Plus kit used for RNA extraction was provided by SparkJade (Jinan, China). Commercially available kits for blood indicator-related tests are provided by Nanjing Jiancheng Institute of Biological Engineering, Nanjing, China. All other chemicals used were of analytical grade.

### 2.2. Fabrication of the Nanocomplex

The wall material ZH was derived from the three-hour hydrolysis product of Zein. Briefly, zein suspension (3% *w/v*) was hydrolyzed by Alcalase (2% *w/w*) at 50 °C, pH 9.0 for 3 h. At the end of hydrolysis, the pH was lowered to 7.0 and the enzyme was inactivated by heating at 95 °C for 5 min. The hydrolysis product was centrifuged at 4000× *g* for 20 min at 25 °C to remove insoluble material and then dialyzed in a dialysis bag (100 Da cutoff) against deionized water for 24 h. Finally, it was freeze-dried and stored at room temperature. FX encapsulated with ZH nanocomplex (FZNP) was fabricated according to the approach of Zhang et al. (2022) [24]. Briefly, ZH (5 g) freeze-dried powder was dissolved in deionized water (1 L) and stirred continuously for 1 h using a magnetic stirrer (800 rpm). FX (200 mg) was dissolved in 10 mL ethanol, which was then added to ZH solution dropwise while keeping the magnetic stirrer at 800 rpm and continue stirring for 1 h. Finally, ethanol was removed from the nanocomplex dispersion using a rotary evaporator (40 °C, −0.1 MPa) and the pH was adjusted to 7.5, followed by centrifugation at 4000× *g* for 20 min at 25 °C to remove insoluble material and freeze-dried to obtain FZNP.

### 2.3. Characterization of FZNP

Freshly prepared samples were diluted 10-fold with deionized water and particle size, PDI and ζ-potential were measured by dynamic light scattering (DLS) analysis (Nano-ZS90, Malvern Instruments, Ltd., Malvern, UK). Encapsulation efficiency (EE) was measured by HPLC with a slight modification of the reference Li et al. (2018) [25], which is set as follows. SB-C18 column, 450 nm wavelength, 0.8 mL/min elution flow rate, 25 °C column temperature, 90% acetonitrile and 10% water as mobile phases, injection volume of 10 μL. A total of 1 mL of freshly prepared dispersion (prior to freeze-drying) was added to 1 mL of MTBE/ acetone (1:1, *v*/*v*) and centrifuged at 4000× *g* for 10 min to collect the organic phase containing FX. Repeat the above steps 2–3 times until the organic phase is clear. The combined organic phases were dried under a stream of nitrogen and added 1 mL of acetonitrile for HPLC test. The EE of FX was determined using the following equation.
EE (%) = FX _(in nanoparticles)_ / FX _(input)_ × 100%

The functional groups of FZNP, ZH and free FX were observed by Fourier Transform Infrared Spectroscopy (FT-IR, Nicolet iS10, Thermo Fisher, Franklin, MA, USA) to determine whether chemical bonds were formed between FX and the wall ZH. Briefly, the samples were mixed with potassium bromide, ground in an agate mortar and pressed into thin slices for the determination, and pure potassium bromide FT-IR was set for all sample backgrounds with the wave number range set to 4000 to 400 cm^−1^.

The physical states of the three samples could be structurally characterized with an X-ray diffraction equipped with Cu Kα radiation (40 kV, 40 mA, D8, Bruker, Bremen, Germany). The diffraction pattern was scanned at a rate of 0.2°/min over a range of 5–50°.

Transmission electron microscopy (TEM, 2100Plus, Jeol, Tokyo, Japan) was used to observe the morphology of the samples. The samples were prepared as follows [25]. The FX and FZNP solutions were each dropped on the carbon film-supported copper grids, dried for one hour and then the samples were negatively stained with 1% (*w/v*) phosphotungstic acid and dried at room temperature. Finally, they were observed under TEM.

### 2.4. In Vivo Experiment

#### 2.4.1. Animals and Experimental Protocols

Six-week-old male C57BL/6 mice were purchased from Weishanglituo Technology Co., Ltd. (Beijing, China). Housing conditions were 24 ± 2 °C, 12 h/12 h light/dark cycle, 55 ± 10% humidity and free access to food and drinking water. After one week of acclimatization, animals were randomly assigned for the first time to a control (NC) (*n* = 8), fed a low-fat diet (LFD: 4% fat, 51% carbohydrate and 20% protein) and a high-fat diet group (HFD, 35% fat, 26% carbohydrate and 26% protein, *n* = 60). Six weeks after induction of the high-fat diet, mice in the HFD group were administered fasting intraperitoneal injections of streptozotocin (STZ, 45 mg/kg, dissolved in 0.1 M sodium citrate buffer) once a week for four weeks. Fasting blood glucose (FBG) was measured, and mice that maintained FBG ≥ 11.5 mM/L for two consecutive weeks were used as diabetic mice (*n* = 40), while the other mice were kept in reserve. Diabetic mice were randomized a second time: diabetic control (db, *n* = 8), positive control (db_MF, *n* = 8), FZNP (*n* = 8), db_FX (*n* = 8), and db_ZH (*n* = 8) groups, respectively. Metformin hydrochloride (MF, 200 μg/g mouse), FZNP (1.2 mg/g mouse, where FX: ZH = 1:29), free FX (40 μg/g mouse) and ZH (1.16 mg/g mouse) were administered to the treatment group (db_MF, db_FZNP, db _FX and db_ZH respectively) every evening at 5 pm the NC and db received the same volume of drinking water at the same time. Body weight, food/water intake and FBG were recorded weekly during the experiment. All experimental protocols used in this study were approved by the Animal Care and Use Committee (No. SPXY2021030503, Ocean University of China) and the animals followed institutional ethical guidelines.

The glucose tolerance test (OGTT) and insulin tolerance test (ITT) were performed as previously described with minor modifications [26]. Briefly, mice were fasted for 8 h four weeks after receiving the treatment intervention, and all mice were gavaged with a dose of 2.0 g/kg of glucose. Blood samples were taken from the tail end at 0, 30, 60, 120 and 180 min after gavage to measure their blood glucose levels, and then the diet was resumed. The ITT experiment was performed three days after the OGTT, in which 0.75 U/kg of insulin was administered subcutaneously to the fasted mice and blood glucose levels were monitored at 0, 30, 60, 120 and 180 min after insulin injection. Data were processed using the trapezoidal rule to estimate the area under the curve (AUC) for OGTT and ITT.

#### 2.4.2. Biochemical Tests in Serum and Tissue

Serum was tested for triglycerides (TG), total cholesterol (T-Cho), high-density lipoprotein (HDL), low-density lipoprotein (LDL), alanine aminotransferase (ALT), aspartate aminotransferase (AST), alkaline phosphatase (AKP) and diabetes-related indicators glucagon-like peptide-1 (GLP-1), fasting serum insulin (INS) and adiponectin (ADPN), and liver glycogen levels. Insulin resistance (IR) plays an important role in the development of T2DM and refers to the reduced ability of target tissues (liver, skeletal muscle and adipocytes) to respond to insulin, a key detrimental factor in the pathogenesis of T2DM. It was calculated using the homeostatic model assessment, the index of insulin resistance (HOMA-IR) and beta cell function (HOMA β-cell), which were estimated by using the equation, as follows:
HOMA-IR = FIN (mIU/L)/FBG (mM/L)/22.5
HOMA-β = 20 × Insulin (mIU/L)/FBG (mM/L) − 3.5

The ground liver tissue is tested for glutathione peroxidase (GSH-Px) activity, superoxide dismutase (SOD) activity, catalase (CAT) activity, non-esterified free fatty acids (NEFA) and malondialdehyde (MDA) by using commercially available kits (Nanjing Jiancheng Institute of Biological Engineering, Nanjing, China). All test procedures and the calculation of results were carried out exactly, according to the instructions.

#### 2.4.3. Histology

Histological analysis of liver was performed by Haematoxylin and Eosin (H&E) staining methods. Mouse liver tissue was fixed in 4% paraformaldehyde and treated with dehydration and decolourization, which was then embedded in paraffin and cut to 5 μm thickness for H&E staining. The stained samples were finally examined under a microscope (Olympus CKX53, Tokyo, Japan).

Pancreatic samples were examined using an immunofluorescence assay. Briefly, pancreas samples were wrapped in 5-μm thick paraffin wax, then dewaxed using ethanol and soaked in a solution of PBS (pH 7.4) containing 3% BSA for 30 min before being incubated with anti-insulin antibody (1:150 *v*/*v*) overnight at 4 °C. Sections were washed three times and then incubated with a secondary antibody sheep anti-rabbit IgG/Cy3 (1:100 *v*/*v*) for 1 h in the dark, and finally stained with DAPI after washing. Images were then taken using a laser confocal microscope (Nikon A1R HD25) and fluorescence intensity was analyzed using ImageJ software (version 1.53K).

#### 2.4.4. Real-Time Quantitative Reverse Transcription Polymerase Chain Reaction Assay (Real-Time RT-qPCR)

Quantification of mRNA expression levels was performed by RT-qPCR assay. First, 100 mg of liver tissue was treated with 1 mL of TRIzol (Invitrogen, Carlsbad, CA, USA). mRNA was extracted by SPARKscript II RT Plus kit (SparkJade, Jinan, China) and quantified by NanoDrop (NanoDrop Technologies, Thermo Fisher, Franklin, MA, USA). Finally, 1 μg of RNA was reversed to cDNA and the corresponding mRNA expression levels were quantified by RT-qPCR detection with specific primers (Table 1).

#### 2.4.5. Western Blot Assay

A sample of 100 mg mouse pancreas tissue was homogenized in RIPA lysate, which was then centrifuged at 12,000× rpm for 20 min (4 °C) to obtain pure proteins for the Western Blot assay. Proteins were quantified and then boiled in loading buffer, followed by electrophoresis of the samples using 10% SDS-PAGE. They were transferred to nitrocellulose using semi-dry blotting [27], which was immersed in 3% skim milk for 1 h. Next, the nitrocellulose was probed with primary antibodies of GNAs, CaMK and β-Actin overnight at 4 °C. The second day, it was washed three times with Tris-buffered saline Tween-20 (TBST) lotion and probed with peroxidase-conjugated secondary antibodies goat anti-rabbit. In the end, the blots were imaged by the Tanon 5200 Imaging System (Tanon, Shanghai, China) after using enhanced chemiluminescence reagents. The grayscale and intensity of the bands were analyzed by ImageJ (version 1.53K).

### 2.5. Statistical Analysis

Statistical analysis was performed using SPSS 23.0. The data normality and homogeneity of variance were analyzed according to the Shapiro-Wilk test and Levene’s test, respectively (*p* > 0.05). ANOVA was used to analyze the main effects between groups, and paired comparisons (Bonferroni adjustment) were used to examine significant differences between groups. 

## 3. Results and Discussion

### 3.1. Characteristic of the FZNP

The results for particle size, zeta potential and PDI were shown in Figure 1a,b. The particle size of FZNP were in the range of 100–200 nm, ZH was near 300 nm, while the free FX particle size values were in the range of the micrometer. The results of their corresponding zeta potential and PDI demonstrated that FZNP had a high negative charge and its PDI value was the lowest. The value of the zeta potential can be used to assess the stability of a colloidal system. Generally, a colloid is considered stable if the absolute value of the zeta potential is higher than 30 mV [28]. The zeta potential of FZNP was −44.03 (±1.58), while the free FX potential was −18.57 (±0.91) and the wall material ZH was −33.63 (±1.42). This indicated that there could be forces of interaction between the ZH and the core FX rather than a simple physical mixing, which allowed the negatively charged ZH to bind more tightly around FX to enhance the stability of the system. This finding was consistent with the following FT-IR results. The low PDI indicated that the particles were uniformly distributed in the solution. The EE of FX encapsulated by this method was 86.14% (±4.76%), and FX:ZH = 1:29 in FZNP was obtained by calculation, whose ratio was used to calculate the gavage amounts of FZNP, FX and ZH in the mouse experiments.

The experimental results of FT-IR were consistent with the previous study [25]. Briefly, FT-IR of the free FX and ZH demonstrated their respective characteristic peaks in agreement with those reported (Figure 1c). FX had a small and broad O-H stretch at 3461 cm^−1^ and a strong hydrophobic C-H stretching vibration at 2923 cm^−1^ as well as C=O stretching (1647 cm^−1^), C-H shearing and bending (1455 cm^−1^), C-O stretching (1031 cm^−1^), etc. Of particular interest was the allenic bond at 1929 cm^−1^, a unique peak for FX that revealed its instability. [25,29]. The characteristic peaks of ZH were: the hydrophilic O-H stretch of ZH (3303 cm^−1^), the C-H stretch representing its hydrophobicity (2924 cm^−1^), and the amide band (1651 cm^−1^) that was still detectable even after hydrolysis. However, after the formation of FZNP, some significant changes in the characteristic peaks of FZNP were observed: firstly, the O-H stretch became stronger and wider, while the hydrophobic C-H stretch at 2927 cm^−1^ decreased in intensity, which could be attributed to the hydrophobic functional group of FX bound to ZH, leading to an increase in the hydrophilic property of FZNP. Secondly, the disappearance of the specific isoprene bond of FX (1929 cm^−1^) was probably due to the entry of FX into the cavity formed by ZH, and the characteristic peak of FX could not be detected by FT-IR, which can also be supported by comparing the spectral results of FX and FZNP between 400 and 1000 cm^−1^. The peaks of FX in this region were known as the fingerprint region, who was unique to FX, and their disappearance also suggested that FX was encapsulated by ZH and existed stably through the interactions between hydrogen bonding, hydrophobicity and electrostatics, etc. X-ray diffraction analysis was performed to clarify the changes in the crystal structure of FX before and after encapsulation. Figure 1d showed the patterns of FX, FZNP and ZH. The strong peaks between 10° and 20° were the characteristic crystalline peaks of FX [30]. For the encapsulated FZNP, most of its characteristic peaks disappeared, which implied that FX is completely amorphous in nanoparticles. The reduction or disappearance of these peaks suggested that the crystalline shape of FX was altered by binding to ZH. Such changes might be due to the existence of interactions between FX and ZH molecules, which was consistent with the results of FT-IR. 

The results of TEM visualized the difference in FX before and after encapsulation. Nanoparticles with diameters between 100–200 nm could be clearly observed in the image of FZNP in a round shape in Figure 1e, which matched the particle size measured by DLS. Meanwhile, the free state of FX produced a large amount of aggregation due to it being carotenoid with mostly hydrophobic groups (Figure 1f). Based on the experimental results of FT-IR and X-ray diffraction, it could be confirmed that FX was encapsulated inside ZH rather than physically mixed. The shaded area in the center of the sphere in Figure 1e should be the FX core, while some of the smaller particles observed could be nanoparticles formed by the self-assembly of short chains of ZH produced after hydrolysis.

### 3.2. Blood Glucose Indicators

C57BL/6 mice experiments were divided into three periods: a period of acclimatization during 6 weeks of feeding HFD, a period of STZ during four consecutive injections of STZ to form T2DM mice and a 4-week period of gavage intervention (Figure 2a). The food intake and FBG assay of mice in the three periods are shown in Figure 1b,c. Throughout the experimental period, the average daily food intake of NC mice was significantly higher than that of the other groups. This was mainly due to differences in food, with mice in the NC group consuming a normal diet (LFD) and T2DM mice groups having HFD. There was no significant difference between the dietary intake of the groups of T2DM mice during the acclimatization and STZ periods. However, during intervention, mice with MF, FX and ZH treatment had significantly higher intake relative to the db group and no difference was observed between the db_FZNP and db. Although there was no difference in food intake between mice in the db_FZNP and db groups, there was a significant difference in their FBG values. As shown in Figure 1c, during the first week of Intervention, the FBG values of all groups injected with STZ did not differ, while the FBG values of mice in the NC group were stable around 8 mmol/L throughout the intervention. During the last week of Intervention, the FBG values in each group were: db 15.81 ± 1.25 mmol/L, db_MF 12.09 ± 0.76 mmol/L, db_FZNP 11.12 ± 0.90 mmol/L, db_FX 12.57 ± 0.86 mmol/L, db_ZH 13.11 ± 0.73 mmol/L, with a significant decrease in FBG values in the db_FZNP and db_MF mice compared to the db group. Only the db_FZNP was not significantly different from the NC group (Bonferroni, *p* = 0.137), i.e., the FBG values of the FZNP-treated mice returned to normal levels, and this decrease in blood glucose was not caused by a decrease in food intake. 

OGTT and ITT are usually used as the main parameters for determining glucose metabolism and pancreatic β-cell function, and can be used as the degree of glucose intolerance and insulin resistance in the organism. Blood glucose levels in the T2DM mouse group increased rapidly within 30 min after sugar administration and then decreased slowly to varying degrees over 180 min (Figure 2e). The intervention of MF, FZNP and ZH resulted in the AUC_ of T2DM mice OGTT was significantly lower than that of mice in the db group (Figure 2d), and they could suppress the elevation of blood glucose levels and alleviate the symptoms of glucose intolerance. Insulin sensitivity was significantly improved in all the intervened T2DM mice (Figure 2f,g), and their AUC_ITT values were all significantly lower than that of the mice in the db group. However, comparing the values of AUC_OGTT and AUC_ITT in the T2DM group with those in the NC group, we found that both of them were higher than those in the NC group, which means that in terms of alleviating glucose intolerance and insulin resistance, neither of the intervention treatments could restore them to normal levels.

### 3.3. Biochemical Parameters of T2DM Mice

There is a significant association between lipid indices (T-Cho, TG, HDL and LDL) and the incidence of T2DM, as T2DM affects the lipid metabolism of the organism [31]. In general, normal organisms had higher HDL and lower TG, T-Cho and LDL levels relative to the T2DM group, which is consistent with the mean values of NC and db in our results (Figure 3a). FX, with and without encapsulation as well as MF treatment (a common drug used for the treatment of T2DM), demonstrated poor results in detecting lipid indices, i.e., they significantly increased serum LDL as well as T-Cho and TG values. The cause of this phenomenon could possibly relate to the stress response of the organism. The intervention of the treatments caused a stress response in the mice, which led to changes in the output of lipid indices in serum. This result was also supported by the H&E histology. Both vacuole and hepatic sinusoid formation of hepatocytes were significantly reduced in the db_FX and db_FZNP compared to the db (Figure 3c). Thus, the function of FZNP and FX in protecting the liver from damage in T2DM mice was verified here. ADPN is a cytokine secreted by adipocytes. By binding to the receptors on target cell membranes, it promotes fatty acid oxidation and glucose uptake and participates in the regulation of glucose and lipid metabolism, exerting antidiabetic and insulin-sensitive effects [32]. As shown in Figure 4, the mean values of INS and glycogen were higher in T2DM mice than those of the normal group NC, while ADPN and GLP-1 values were lower than those of NC, but only INS values were significantly different. After the intervention of four treatments, INS values were significantly lower in all treatment groups compared to the db. In the ADPN assay, FZNP and MF treatments significantly elevated it relative to db, while wall material ZH and free FX had no significant effect on ADPN. 

HOMA-IR and HOMA-β cell levels were used to assess insulin resistance (IR) and islet β-cell function. HOMA-IR levels were significantly higher and HOMA-β cell levels were significantly lower in the db group compared to the NC group, suggesting that a long-term high-fat diet and STZ injection led to an increased occurrence of IR and a decrease in their islet β-cell function in mice (Figure 4e,f). After gavage intervention, HOMA-IR was significantly lower in FZNP-treated mice compared to db and did not differ from the NC group, which was superior to the db_MF. Notably, the HOMA-β cell assay demonstrated that the FZNP treatment was the only one that was not significantly different from NC, and all other groups of mice had a lower HOMA-β cell than NC. This means that FZNP restored the HOMA-β cell values of T2DM mice to normal levels, while neither free FX nor metformin hydrochloride achieved this efficacy during the experimental period.

High-fat diets usually lead to an overproduction of reactive oxygen species (ROS) [33], which is responsible for cellular damage. According to the chemical structure of FX, it has an epoxide group and hydroxyl group; therefore, FX is often used as a strong antioxidant [34], whose supplementation reduces oxidative stress [35]. In the study, some indicators of the antioxidant capacity, such as GSH-PX, SOD and CAT, were significantly higher in the group treated with FZNP than the other groups (Figure 4c). In contrast, in the db_FX group, their mean values were higher than those of the NC and db groups, but there was no significant difference, which then affirmed that the encapsulated FZNP could better inhibit the excessive production of ROS and thus avoid cell damage. NEFA is highly cytotoxic; it can damage cell membranes, mitochondria and cause many diseases [36]. In mice with T2DM, the inhibitory effect of insulin on NEFA is diminished and therefore the level of NEFA is usually increased [2]. MDA usually reflects the severity of free radical attack on body cells, while SOD reflects the body’s ability to clear oxygen radicals [37]. Normally they have opposite results, which is consistent with the results of the present study. NEFA demonstrated an increasing trend in db (Figure 4d), i.e., the diabetic mice without any intervention, while the mean values of NEFA in db_FZNP, db_FX, and db_ZH groups were lower but not significant. However, in the results of MDA, a significant decrease in MDA values was observed in the db_FZNP and db_MF groups. This was most likely due to a higher sensitivity of MDA, while NEFA only demonstrated a trend during the 4-week intervention. 

### 3.4. Mechanisms Affecting Diabetes

AKT, AMPK, ClUT2, INSR and PI3K in glycogen synthesis and gluconeogenesis pathways are key genes controlling glucose transport, glucose metabolism and glucose homeostasis [38,39]. To further validate the treatment group effect, the RT-qPCR technique was used to detect the gene expression of AKT, AMPK, GLUT2, INSR and PI3K in mouse liver using β-Actin and GAPDH as an internal reference, respectively. Compared with the NC group of normal mice, the expression of all the above genes was reduced in db mice, while FZNP and FX significantly improved the expression of the above genes, i.e., the above genes were differentially up-regulated by FZNP and FX treatment (Figure 4g). Both FZNP and FX might inhibit the abnormal insulin signaling pathway, improve glucose uptake and hepatic glucose metabolism and thus attenuate insulin resistance.

The superior effect of FZNP over FX for the regulation of GLUT2 gene may be due to the protection of wall material ZH avoiding catabolism in the gastric phase and reaching directly to the intestinal phase to be utilized by the organism, which was confirmed in the in vitro digestion simulation experiments that we did previously [24]. The release rate of FX from FZNP in the gastric phase was 0, versus 13.19% (±5.78%) for the free FX in the gastric phase. This result was also consistent with the data on glycogen and GLP-1, i.e., the mean values of glycogen and GLP-1 in the serum of FZNP-treated mice were increased compared to db. Due to the up regulatory effect of FZNP on the gene expression levels of the PI3K/AKT signaling pathway, it could lead to changes in glucose uptake and hepatic glucose metabolism [40].

Insulin is known to be secreted by β-cells in the pancreas and is one of the most important metabolic hormones in the control of blood glucose [41]. The expression of insulin in the pancreas can be measured using immunofluorescence (ICC) labeling of the insulin in the pancreas (pink fluorescent area in Figure 5a). It clearly demonstrated that mice in the db group exhibited a significant reduction in insulin secretion significantly (in Figure 5a,b). After administration of the intervention, FZNP and FX significantly increased insulin secretion, which was consistent with the ITT results (Figure 1). Notably, there was no difference between db_FZNP and db_FX and was also not significantly different from the NC group, with a better effect than the db_MF group. In other words, the function of mouse pancreatic β-cells was restored after FX and FZNP interventions. Glucose entered the pancreatic β-cells and hepatocytes through the membrane transporter GLUT2 and stimulated the expression of CaMK and GNAs proteins in pancreatic β-cells. Due to their low expression of GLUT2 in T2DM mice (Figure 4g), the expression of GNAs and CaMK detected in WB experiments was significantly lower relative to the NC group (Figure 5d,e). When FZNP and FX intervened, on one hand it increased the expression of CaMK and GNAs proteins (Figure 6, the right part), thus stimulating β-cells to produce more insulin and enhancing the binding of insulin to the receptor (INSR) on the surface of hepatocytes, and stimulating hepatocytes in cooperation with glucose. On the other hand, they could up-regulate the gene expression of GLUT2, which contributed to the transport of glucose and in turn up-regulated PI3K expression. The up-regulation of PI3K/AKT pathway stimulates hepatocytes to convert glucose into hepatic glycogen and store it, resulting in a hypoglycemic effect (Figure 6, the left part).

## 4. Conclusions

In conclusion, this study used a mono-carrier wall to encapsulate FX to form a nanocomplex with stable structure and chemical properties. Its efficacy in alleviating diabetes in mice was evaluated by in vivo experiments. Both FZNP and FX significantly repaired the damage to liver cells and pancreatic islet β-cells, increased SOD and decreased INS values in serum, respectively, and upregulated gene expression in the PI3K-AKT pathway, as well as protein expression of CaMK and GNAs in the pancreas. Moreover, FZNP also showed some significant advantages over free FX, for example, it could more effectively reduce FBG in T2DM mice and restore it to normal levels, and it increased ADPN and GSH-PX values and reduced HOMA-IR and MDA values in serum more significantly. In addition, FZNP up-regulated the expression of the GLUT2 gene, which facilitated the transport of glucose through pancreatic and hepatic cells to accomplish the corresponding glucose metabolism and regulation of glycogen. Therefore, FZNP are promising in alleviating T2DM and can be used as future food, pharmaceutical and material interventions for the treatment of T2DM materials, and this study fills the gap in the in vivo evaluation of mono-carrier encapsulated active substances, providing an important foundation for active substance encapsulation and delivery studies.

## Figures and Tables

**Figure 1 antioxidants-11-01976-f001:**
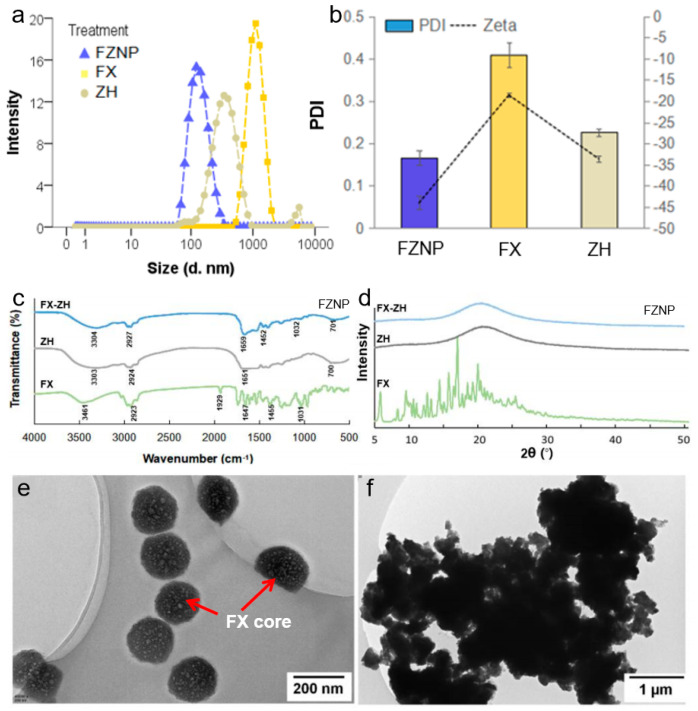
Characteristics of the fucoxanthin encapsulated with zein hydrolysate nanocomplex (FZNP). (**a**) The diameter, (**b**) zeta-potential and PDI, (**c**) Fourier transform infrared spectra and (**d**) X-ray diffraction of FZNP, the wall material zein hydrolysate (ZH) and free fucoxanthin (FX). (**e**) The TEM image of FZNP and (**f**) free FX solution. (**c**–**f**) were cited from the results of Zhang et al. (2022) (Reprinted with permission from Ref. [24]. 2022, Elsevier).

**Figure 2 antioxidants-11-01976-f002:**
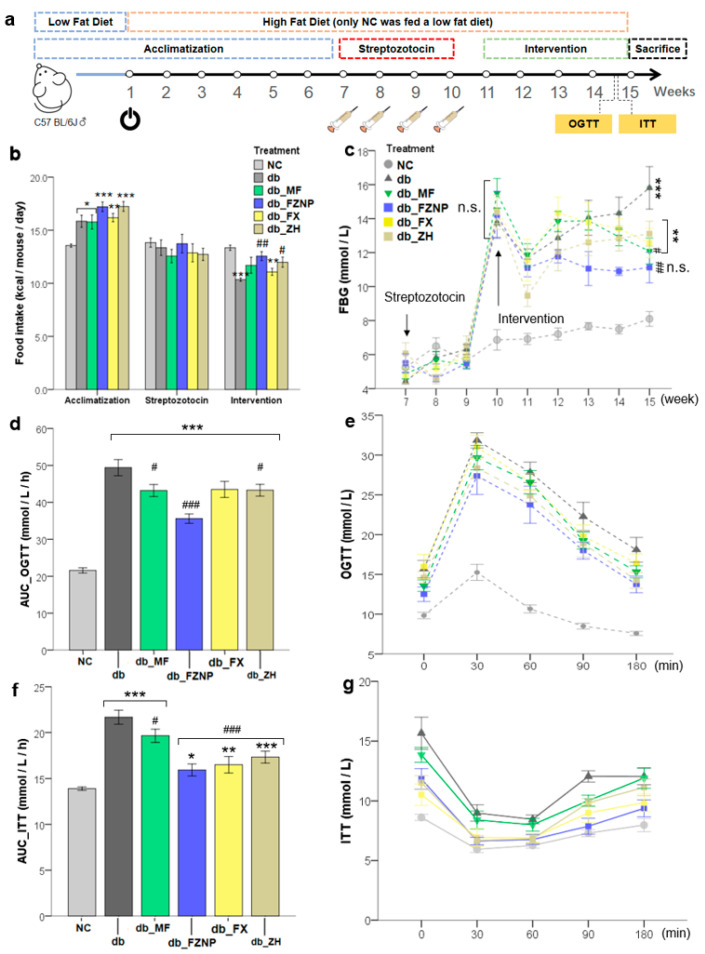
The schedule of mice experiments during 15 weeks (**a**), daily food intake (**b**), fasting blood glucose testing in mice (FBG, **c**), oral glucose tolerance (OGTT, **e**) and insulin tolerance (ITT, **g**) and the area under the curve of glucose (AUC_OGTT, **d**) and insulin (AUC_ITT, **f**). The asterisk including * (*p <* 0.05), ** (*p* ≤ 0.01), and *** (*p* ≤ 0.001) indicate the significant difference compared with the NC group. The hashes including # (*p* < 0.05), ## (*p* ≤ 0.01) and ### (*p* ≤ 0.001) indicate the significant difference from the db group.

**Figure 3 antioxidants-11-01976-f003:**
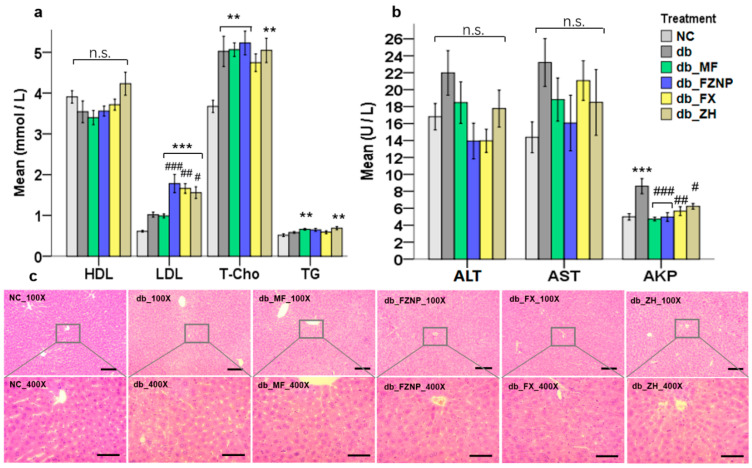
Biochemical tests in serum and Histology of Hepatocyte. The level of HDL, LDL, T-Cho and TG in the serum of mice (**a**). The value of ATL, AST and AKP in serum (**b**). The asterisk including ** (*p* ≤ 0.01), and *** (*p* ≤ 0.001) indicate the significant difference compared with the NC group. The hashes including # (*p <* 0.05), ## (*p* ≤ 0.01) and ### (*p* ≤ 0.001) indicate the significant difference from the db group. Microscopy images of H&E-stained liver tissues (**c**, 100× scale bar 100 μm, 400× scale bar 50 μm).

**Figure 4 antioxidants-11-01976-f004:**
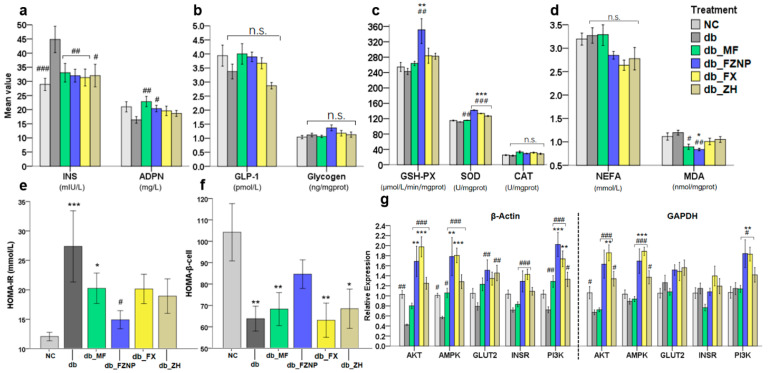
Indicators of blood, insulin and glucose metabolism in serum as well as liver tissue and gene expression measured by qPCR. The levels of INS/ADPN (**a**), GLP-1/Glycogen (**b**), glutathione peroxidase (GSH-Px) activity, superoxide dismutase (SOD) activity and catalase (CAT) activity (**c**), non-esterified free fatty acids (NEFA) and malondialdehyde (MDA) (**d**), HOMA-IR (**e**), and HOMA-β (**f**), expression of AKT, AMPK, GLUT2, INSR and PI3K genes in the liver with β-Actin (right part) and GAPDH (left part) as internal reference (**g**). The asterisk including * (*p <* 0.05), ** (*p* ≤ 0.01), and *** (*p* ≤ 0.001) indicate the significant difference compared with the NC group. The hashes including # (*p <* 0.05), ## (*p* ≤ 0.01) and ### (*p* ≤ 0.001) indicate the significant difference from the db group.

**Figure 5 antioxidants-11-01976-f005:**
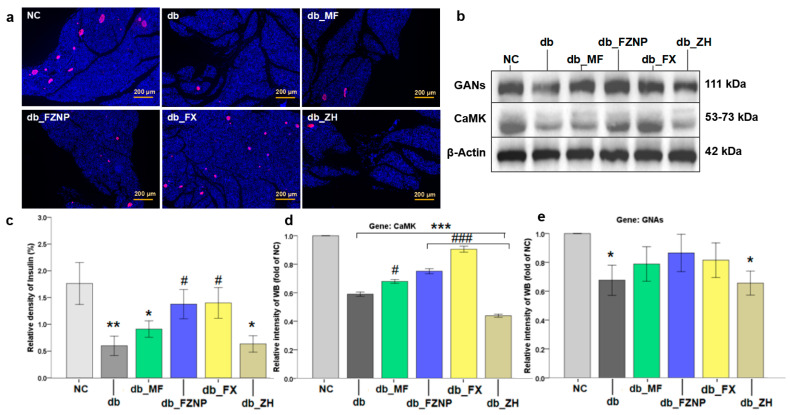
Testing of insulin and protein expression in the pancreas. Pancreatic tissue was analyzed by immunofluorescence. Insulin-positive cells were shown to fluoresce in pink and nuclei were stained with DAPI (blue fluorescence) to observe the fluorescence intensity of insulin-positive areas (**a**). The intensity of each sample was quantified using ImageJ software (**b**). Western blot images of GNAs, CaMK and β actin expression (**c**). The analysis of ImageJ for CaMK (**d**) and GNAs (**e**) expression. The asterisk including * (*p <* 0.05), ** (*p* ≤ 0.01), and *** (*p* ≤ 0.001) indicate the significant difference compared with the NC group. The hashes including # (*p* < 0.05) and ### (*p* ≤ 0.001) indicate the significant difference from the db group.

**Figure 6 antioxidants-11-01976-f006:**
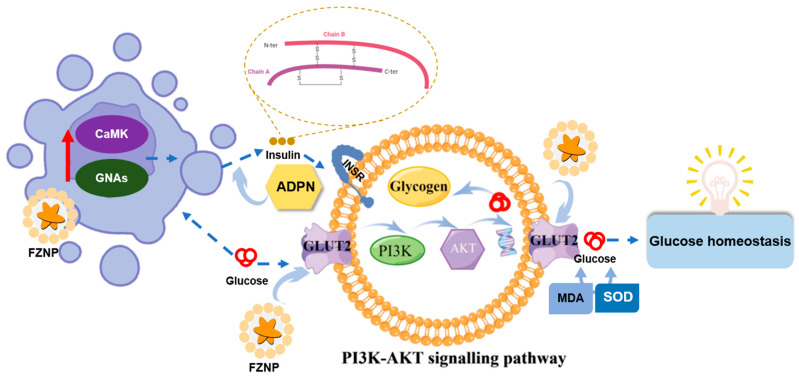
Schematic illustration of the mechanism of the regulation of CaMK and GNAs expression and the PI3K-AKT signaling pathway by encapsulated fucoxanthin nanocomplex (FZNP).

**Table 1 antioxidants-11-01976-t001:** Primer sequences for RT-qPCR detection.

Gene	Sequence	
* β-Actin	Forward	5′-GTGCTATGTTGCTCTAGACTTCG-3′
	Reverse	5′-ATGCCACAGGATTCCATACC-3′
INSR	Forward	5′-CGTGTTGCGGTTAAGACTGTCAATG-3′
	Reverse	5′-CCAAGAAGGCGGACCACATGATG-3′
PI3K	Forward	5′-GCTTTGCCGAGCCCTACAAC-3′
	Reverse	5′-GTCATTGTGCTGCACGAGGG-3′
AKT	Forward	5′-TCAGGATGTGGATCAGCGAGAGTC-3′
	Reverse	5′-AGGCAGCGGATGATAAAGGTGTTG-3′
AMPK	Forward	5′-CGAGTGTTCGGAGGAGGAGGTC-3′
	Reverse	5′-GTGGGCTGGTTGCTAGGTAGAAATC-3′
GLUT2	Forward	5′-ACAGTCACACCAGCATACACAACAC-3′
	Reverse	5′-CCGAGCCACCCACCAAAGAATG-3′

* β-Actin were employed as an internal reference.

## Data Availability

The data presented in this study are available on request from the corresponding author.

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
