# Peer review of "In Vivo Assessment of the Effects of Mono-Carrier Encapsulated Fucoxanthin Nanoparticles on Type 2 Diabetic C57 Mice and Their Oxidative Stress"

_antioxidants, 2022, doi:10.3390/antiox11101976_

Round 1
Reviewer 1 Report
The manuscript deals with the development of monolayer-encapsulated fucoxanthin nanoparticles and assessment of the developed formulation on type 2 diabetic C57 mice. the manuscript represents an interesting topic and the work is well characterized; however, some concerns have to be addressed prior to considering publishing:
1. Introduction: lines 86-90: questions are not appropriate in this part. Mentioning the goal of the study in direct simple words will be more professional and clearer to the reader.
2. Methodology: Lines 119-128: the method of HLC used for determination of drug should be validated in terms of accuracy and precision. If the method is adopted from a previous work, reference should be mentioned.
3. Results and discussion: Line 244: the high negtive charge should be explained.
4. Lines 279-285: the core of the drug described by the authors is nor clear in the image. The uthor should better desribe the spherical shape of the nanoparticles that could indicate encapsulation of the drug within it. On the other side, no particles are formed in the drug solution.
Reviewer 2 Report
The manuscript of Zhang et al. entitled "In vivo assessment of the effects of monolayer-encapsulated fu-2 fucoxanthin nanoparticles on type 2 diabetic C57 mice and their 3 oxidative stress” describes a novel therapy against type 2 diabetes based on fucoxanthin-loaded Zein nanoparticles. The manuscript is well written, the topic is original and the interest to a general audience. Although some interesting results have been presented, several questions need to be addressed before this manuscript could be considered for publication. I would recommend a resubmission with minor revision based on the following comments:
- Line 12. There is two times “from marine origin”.
- From a synthetic point of view, there are inconsistencies in FX encapsulation amount. In the cited reference (J. Industrial and Engineering Chemistry Volume 114, 25 October 2022, Pages 96-107), FX is encapsulated at 100 µg/10 mL concentration with encapsulation yield of 86%. In the present manuscript, following the same protocol of synthesis, FX is encapsulated in a much higher concentration (200 mg/mL) with similar encapsulation yield (86.14%). Please can the authors clarify this point.
- From a morphological point of view, there is not any evidence in the manuscript that the FZNP have core-shell structure. From the TEM images it is not possible to distinguish the layer of the wall that the authors report. The authors most likely have solid nanospheres filled with FX and Zein protein.
- Line 477. There is no evidence of any monolayer wall. I would recommend suppressing the words “monolayer wall”.
- In general, there is overlapping in Figures with a preview work of the authors. Figures 1c, 1d and 1f are practically the same as the article J. Industrial and Engineering Chemistry Volume 114, 25 October 2022, Pages 96-107. Moreover, Figure 1f does not show any relevant information. I would recommend suppressing it.
Round 2
Reviewer 1 Report
The authors have addressed the reviewer comment in comprehensive way. The paper can be accepted in its current form.